# Roto-Chemical Heating with Fall-Back Disk Accretion in the Neutron Stars Containing Quark Matter

**Wei Wei \*** 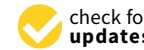**, Xin-Yu Xu, Kai-Tuo Wang and Xiao-Hang Ma**

Huazhong Agricultural University, Wuhan 430070, China; xuxinyu.xxy@webmail.hzau.edu.cn (X.-Y.X.);
katalang@webmail.hzau.edu.cn (K.-T.W.); mxh.hzau.edu.cn@webmail.hzau.edu.cn (X.-H.M.)
\* Correspondence: weiwei1981@mail.hzau.edu.cn

**Abstract:** Probing quark matter is one of the important tasks in the studies of neutron stars (NS). Some works explicitly consider the existence of quark matter in the appearance of hybrid star (HS) or pure quark star (QS). In the present work, we study the roto-chemical heating with accretion in HS and QS, and compare their chemical evolution and cooling features with pure NS. Different from HS and NS, there are two jumps in the chemical evolution of QS, which results from the fast direct Urca (Durca) reaction causing the fast recovery to chemical balance. However, the sudden change in the chemical evolution doesn't provide an obvious heating effect in the thermal evolution. Differently, the roto-chemical heating effect appears both in the accretion phase and spin-down phase of the HS, and the heating platform in the accretion phase relies on the accretion rate. Larger accretion rate results in larger chemical deviation, higher and longer heating platform, and earlier appearance of the heating effect. Interestingly, with the disappearance of the heating effect in the accretion phase, the surface temperature drops fast, which is another possibility of the rapid cooling trend of the NS in Cas A. Furthermore, the surface temperature of the QS is obviously lower than the HS and NS, which is a latent candidate for the explanation of the old classical pulsar J2144-3933 with the lowest known surface temperature.

**Keywords:** neutron star; quark matter; roto-chemical heating; thermal evolution

---

## 1. Introduction

Neutron star (NS) cooling is one of the important tools to explore dense matter and its inner structure [1–4]. NS loses thermal energy by neutrino emission initially, which is replaced by surface photon radiation after $10^5$ yr . In standard cooling models, NS cools to surface temperature $\sim 10^4$ K between $10^6 \sim 10^7$ yr [5]. However, some pulsars with higher surface temperature have been observed, such as millisecond PSR J0437-4715, J2124-3385 ($\sim 10^5$ K) [6,7] and classical PSR B0950+08($1 \sim 3 \times 10^5$ K) [8]. Then, a couple of heating mechanisms have been extensively studied, among which roto-chemical heating [9–11] and vortex creep heating [10,12–14] are important for old millisecond pulsars and classical pulsars. Furthermore, roto-chemical heating with fall-back disk accretion was discussed [15], which is important in the explanation of classical pulsars with high temperature and solves the problem that the classical pulsar's temperature explained by roto-chemical heating needs an implausibly short initial rotation period ($P_0 \le 10$ ms) [8].

Roto-chemical heating results from the weak reaction deviation caused by the spin-down (or up) compression (or expansion). If the NS accretes from a supernova (SN) fall-back disk, the accretion process affects the thermal evolution of the NS by the impact on the spin of the star. The studies indicate that two roto-chemical deviation phases (spin-up phase and spin-down phase) appear in the NS, but the spin-down deviation results in chemical heating only. The cooling curve relies on accretion disk mass (accretion rate), magnetic field decay rate, and initial field strength [15].

Probing quark matter is one of the important tasks in the studies of NS. Some works explicitly consider the existence of quark matter in the appearance of HS or pure QS [16,17]. The cooling process of the NS containing quark matter has an obvious difference from pure NS [18–20]. Recently, the observation of the old classical pulsar J2144-3933 with age $3 \times 10^8$ yr shows the surface (un-redshifted) temperature of $T < 42{,}000$ K, which is reported as the coldest known neutron star, and allows us to study thermal evolution models of old neutron stars. This observation of the temperature was explained by the NS models with traditional roto-chemical heating and considering frictional heating from the motion of neutron vortex lines [21]. QS always has lower surface temperature as the fast direct Urca processes operate in it. The QS model with accretion roto-chemical heating may provide another possibility for the explanation of PSR J2144-3933 observations. This work aims to study the roto-chemical heating effect with fall back disk accretion in the NS containing quark matter, compare their chemical and heating evolution features with pure NS, and try to give some explanations to the thermal emission observations of pulsars. In Section 2, we provide the chemical and thermal evolution equations, consider the fall-back disk accretion process in the star, and provide an application in an uniform star model with different accretion parameters. Section 3 display the results and corresponding explanations. Conclusions and discussions are provided in Section 4.

## 2. Chemical and Thermal Evolution of the QS and HS

For the pure NSs, the relative concentrations of $n$, $p$, and $e$ matter are dominated by weak reactions,

$$p + p + e^- \rightarrow n + p + \nu_e, n + p \rightarrow p + p + e^- + \bar{\nu}_e, \tag{1}$$

$$p + n + e^- \rightarrow n + n + \nu_e, n + n \rightarrow p + n + e^- + \bar{\nu}_e. \tag{2}$$

Equations (1) and (2) are $p$ branch and $n$ branch modified Urca (Murca) reactions operated in the nuclear matter. QSs contain quark matter purely, in which the relative concentrations of $u$, $d$, $s$, and $e$ matter are dominated by weak reactions [1],

$$u + e^- \rightarrow d + \nu_e, d \rightarrow u + e^- + \bar{\nu}_e, \tag{3}$$

$$u + e^- \rightarrow s + \nu_e, s \rightarrow u + e^- + \bar{\nu}_e. \tag{4}$$

Reactions (3) and (4) are direct Urca (Durca) reactions operated in the quark matter. NSs contain quark matter partially always are called HSs, and all reactions discussed above (Equations (1)–(4)) are involved in the relative concentrations of $n$, $p$, $u$, $d$, $s$ and $e$ matter in the HSs.

The stars spin down (or up) and change the chemical potential of each particle species, which results in chemical deviations. The chemical potential differences in the nuclear matter and quark matter are written as [9,22]

$$\begin{aligned}
\delta\mu_N &= \delta\mu_n - \delta\mu_p - \delta\mu_e, \\
\delta\mu_D &= \delta\mu_d - \delta\mu_u - \delta\mu_e, \\
\delta\mu_S &= \delta\mu_s - \delta\mu_u - \delta\mu_e,
\end{aligned} \tag{5}$$

where $\delta\mu_i = \mu_i - \mu_i^{eq}$ is the chemical potential deviation of species $i$ [9,22].

For the pure NS, the evolution equation of the chemical potential deviation is

$$\frac{d\delta\mu_N}{dt} = -E_{xx}(n\frac{E_{nx}}{E_{xx}}\frac{\Omega\dot{\Omega}}{G\rho_c}) - E_{xx}\Gamma_n. \tag{6}$$

The two terms on the right side represent the competition between deviation caused by spin down (or up) compression and the weak reactions, and $\Gamma_n$ is the reaction rate per baryon of Murca reactions of nuclear matter. For the other quantities, we refer to the reference [9].

For the pure QS, the evolution equation of the chemical potential deviation is

$$\frac{d\delta\mu_D}{dt} = \frac{d\delta\mu_S}{dt} = -E_{xx}(n\frac{E_{nx}}{E_{xx}}\frac{\Omega\dot{\Omega}}{G\rho_c}) - E_{xx}(\Gamma_d + \Gamma_s),\tag{7}$$

similarly where $\Gamma_d$ and $\Gamma_s$ are the reaction rate per baryon of Durca reactions of quark matter; other symbols refer to the reference [22].

For the HS, considering the effect of deconfinement phase transition on the chemical deviation process, the chemical evolution equations are [23]

$$\frac{d\delta\mu_D}{dt} = \frac{2\alpha-1}{3}\frac{\mu_d}{n_q}\frac{dn_q}{dt} - \Gamma_d E_{xx}^d(\mu),\tag{8}$$

$$\frac{d\delta\mu_S}{dt} = -\frac{2-\alpha}{3}\frac{\mu_s}{n_q}\frac{dn_q}{dt} + \Gamma_s E_{xx}^s(\mu),\tag{9}$$

$$\frac{d\delta\mu_N}{dt} = [\frac{(3\pi^2 n_n)^{\frac{2}{3}}}{((3\pi^2 n_n)^{\frac{2}{3}}+m_n^2)^{\frac{1}{2}}} - \frac{(3\pi^2 n_p)^{\frac{2}{3}}}{((3\pi^2 n_p)^{\frac{2}{3}}+m_p^2)^{\frac{1}{2}}} \quad -(3\pi^2 n_p)^{\frac{1}{3}}]\frac{1}{3n_h}\frac{dn_h}{dt} - \Gamma_n E_{xx}^n(\mu).\tag{10}$$

where $n_i$ and $m_i$ are the baryon number density and mass of different particle and $\alpha = n_n/n_h$ [23].

To solve these equations, a thermal equation is needed and determined by

$$c_v\frac{dT}{dt} = \sum_i \Gamma_i \delta\mu_i - \sum_i \epsilon_i - \dot{E}_\gamma.\tag{11}$$

Following the previous studies, a fall-back accretion disk forms after the core collapse of the NS progenitor. The accretion process affects the spin, surface magnetic field, particles chemical potential, and cooling process simultaneously. The magnetic field evolution couples to the cooling process through the spin evolution. For the spin and magnetic evolution equations, we refer to the reference [15]. Accretion disk mass $M_d$, initial magnetic strength $B_0$, period $P_0$, magnetic decay parameter $M_0$, and initial disc radius $R_d$ are all accretion model parameters listed in Table 1.

**Table 1.** Model parameters for the initial fall-back disk.

| Model No. | $M_d$ ($M_\odot$) | $R_{d,10}$ ($10^{10}$ cm) | $P_0$ (ms) | $B_0$ ($10^{12}$ G) | $M_0$ ($10^{-5}$ $M_\odot$) |
|---|---|---|---|---|---|
| 1 | 0.28 | 0.1 | 14 | 3 | 1 |
| 2 | 0.01 | 0.1 | 14 | 3 | 1 |
| 3 | 0.28 | 0.1 | 14 | 3 | 10 |
| 4 | 0.28 | 0.1 | 14 | 10 | 1 |

## 3. Results

An application to a uniform star model is considered. The baryon number density is $n = 0.4$ fm$^{-3}$ and $\alpha = 8/9$ for the HS, which is a reasonable average density for neutron stars and ensures reasonable star mass and radius. With this density, the Murca processes operate in nuclear matter and Durca processes operate in quark matter. The thermal structures of the crust and core evolve essentially independently during the epoch of neutrino cooling. In this approximation, the core is assumed to be isothermal and to contain all of the star's mass and thermal energy. The temperature of the core is regulated by the loss of neutrinos which do not interact with the crust. The surrounding crust acts as a thin insulating envelope with no sources or sinks of energy and contains all of the temperature gradient. Thus, for a given core temperature, the heat flux through the crust is constant, and the effective surface temperature is determined entirely by processes within the crust. As a good approximation, one can

treat matter in the crust as a two-component plasma with one species, the ions (consisting of nuclei and some bound electrons), immersed in a uniform neutralizing background of free electrons. No other particles (except photons) are present. We generally assume that, for densities less than $6.6 \times 10^6$ gcm$^{-3}$, the nucleus present is $^{56}Fe$. Superfluidity of neutrons isn't taken into account. With these assumptions, the surface temperature is calculated by using the temperature relation between interior and surface $T_b = 1.288 \times 10^8 (T_{s,6}^4 / g_{s,14})^{0.455}$, where $T_b$ is the temperature at the inner boundary of the envelope, $T_{s,6} = T_s / 10^6$ is the effective surface temperature and $g_{s,14} = g_s / 10^{14}$ is the surface gravity of the neutron star [24].

Quark stars may be bare initially and build up a nuclear crust after accretion. In our calculations, the formation process of the crust isn't considered because the formation of crust is of short time scale, comparing to the long time scale of thermal evolution process. Thus, we assume that the quark stars, hybrid stars, and neutron stars already have the same crust configuration and used the same equation to describe the relation between the interior temperature and surface temperature. It is reasonable for us to compare heating effects in different star models.

Then, we integrate the chemical, thermal, magnetic and spin equations. Figures 1–4 demonstrate surface magnetic strength, spin angular frequency, chemical deviation, and surface temperature of pure NS (black line), pure QS (red line) and HS (blue line) with different parameters as models showed in Table 1. To analyze the effect of the accretion in different compact star models (pure NS, QS and HS) clearly, we show curves of pure NS, HS, and QS in each figure for comparison. The chemical deviation of the HS is represented by the deviation of nuclear matter (Equation (10)). The evolution of spin and surface magnetic field depend on the accretion parameters and accretion process, which aren't related to the inner matter of star. In each figure, accretion processes for three stars are same, so the spin angular frequency and surface magnetic field curves of pure neutron star, hybrid star, and quark star overlap each other. However, the surface temperature and chemical deviation are related to the star model; they have different evolution curves in different star models.

As the previous work on the pure NS showed [15], the accretion process leads to two roto-chemical deviation phases (spin-up phase and spin-down phase), and the cooling curve is strongly dependent on the accretion disk mass (accretion rate), the magnetic decay rate, and initial magnetic field. The effect of accretion process on the chemical deviation and cooling of the HS and pure QS has some similar features as in the pure NS.

Notably, the chemical evolution in the pure QS has some special features different from NS and HS. There are two jumps in the chemical evolution of pure QS, but HS and NS only have much smoother change. The first jump happens at the end of the accretion phase, when the accretion ceases abruptly and the star enters the propeller phase. In addition, the second jump happens at the begin of the radio phase, when the spin up rate $\dot{\Omega}$ changes from positive to negative. The chemical deviation depends on the competition between spin up (or down) and weak reactions. The dominant weak reaction in the pure QS is quark Durca reaction, whose reaction rate is two orders higher than the Murca reaction that happened in the nuclear matter. The much higher weak reaction rate makes the system easier to recover to chemical balance. When the spin up (or down) rate changes abruptly, a large reaction rate makes the system easier to recover to the balance and leads chemical deviation decrease obviously, so we see a sharp change of chemical deviation in the QS like a jump. On the contrary, the much slower Murca reaction operates in the HS and NS, so we see much slower change in the chemical deviation. The chemical deviation in HS is obviously higher than pure NS because the effect of deconfinement is considered in the HS [23].

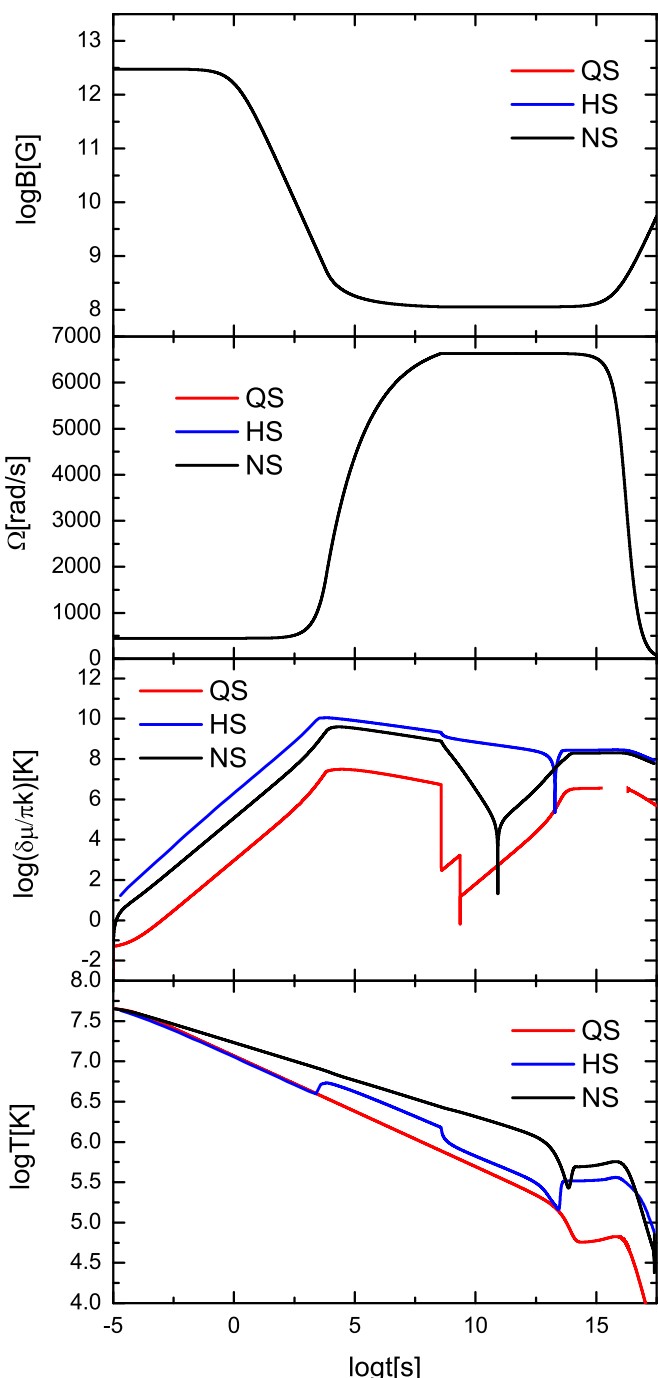

**Figure 1.** Surface magnetic field, spin angular frequency, chemical deviation, and surface temperature as a function of time with parameters of model 1. Pure neutron star, pure quark star, and hybrid star are represented by black line, red line, and blue line, respectively. The units of the temperature, chemical deviation, magnetic field, spin angular frequency, and time are K, K, G, rad/s, and s, respectively.

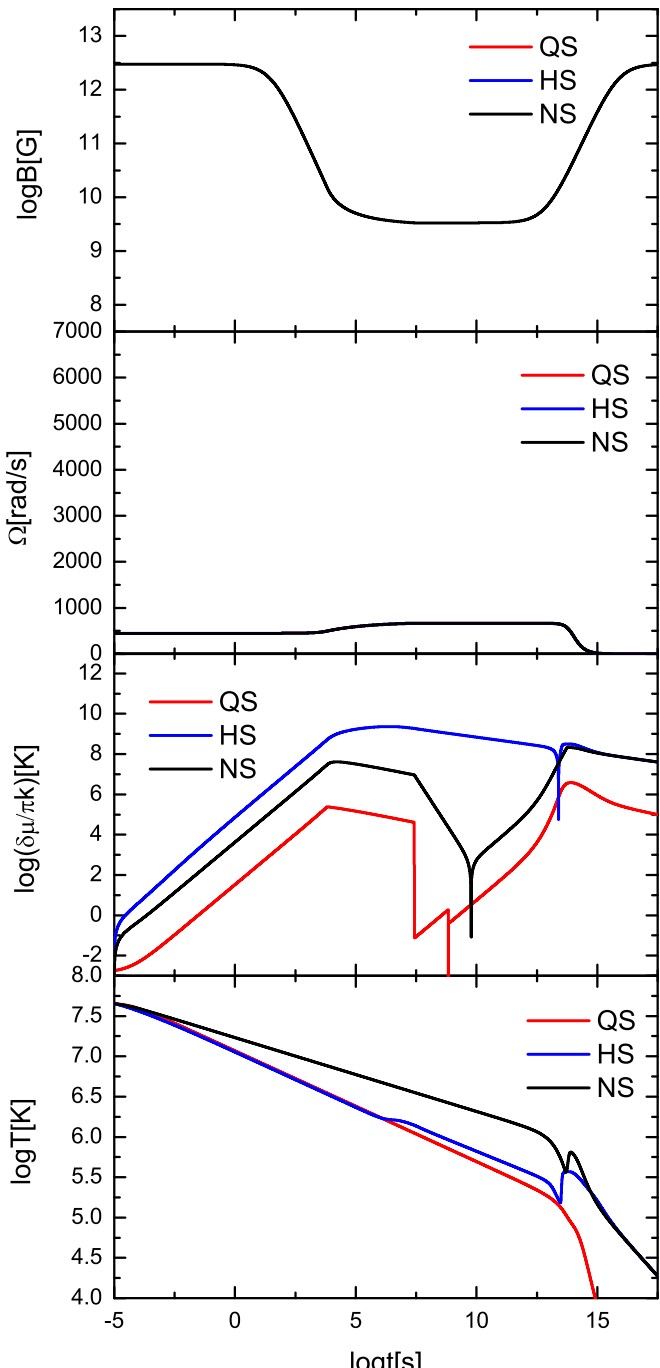

**Figure 2.** Surface magnetic field, spin angular frequency, chemical deviation, and surface temperature as a function of time with parameters of model 2. Pure neutron star, pure quark star, and hybrid star are represented by black line, red line, and blue line, respectively. The units of the temperature, chemical deviation, magnetic field, spin angular frequency, and time are K, K, G, rad/s, and s, respectively.

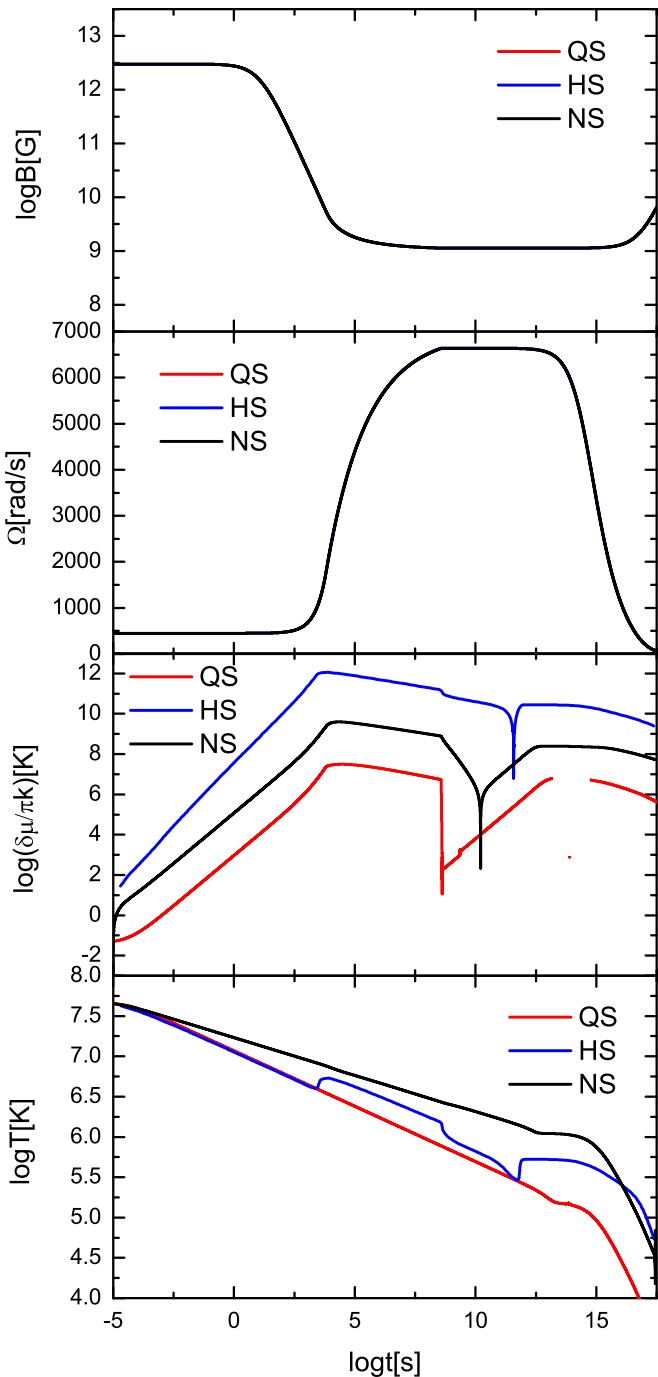

**Figure 3.** Surface magnetic field, spin angular frequency, chemical deviation, and surface temperature as a function of time with parameters of model 3. Pure neutron star, pure quark star, and hybrid star are represented by black line, red line, and blue line, respectively. The units of the temperature, chemical deviation, magnetic field, spin angular frequency, and time are K, K, G, rad/s, and s, respectively.

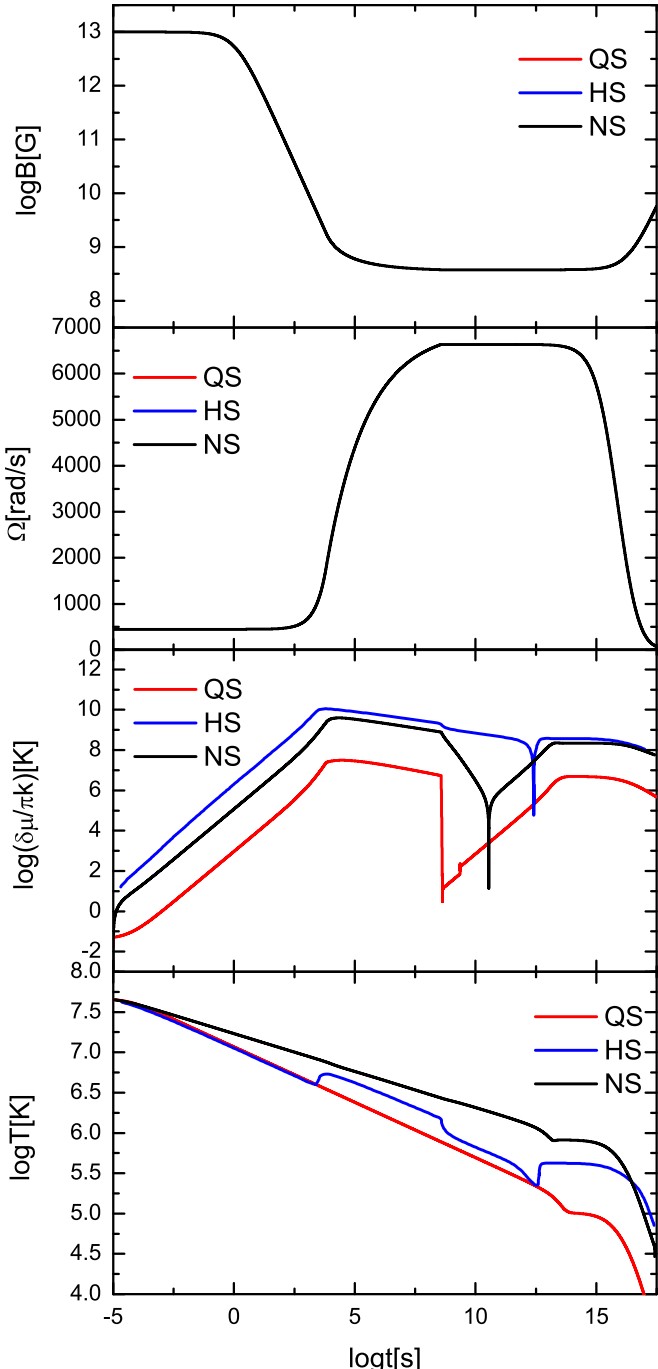

**Figure 4.** Surface magnetic field, spin angular frequency, chemical deviation, and surface temperature as a function of time with parameters of model 4. Pure neutron star, pure quark star, and hybrid star are represented by black line, red line and blue line, respectively. The units of the temperature, chemical deviation, magnetic field, spin angular frequency, and time are K, K, G, rad/s, and s, respectively.

The thermal evolution depends on the heat capacity of the star, weak reactions emissivity, and heating effects operating in the star. The surface temperature of the NS is the highest because the slower nuclear Murca process operates in it; on the contrary, the QS temperature is the lowest because of the larger reaction emissivity of the quark Durca process. For the HS, quark Durca and nuclear Murca both operate in the star, and the thermal evolution is dominated by the quark matter at the early time. After the heating mechanism is strong enough to affect the cooling process, quarks and nucleon co-decide the thermal evolution of the star.



In the previous research of pure NS, the roto-chemical heating effect doesn't appear in the accretion phase, although there is an obvious chemical deviation (spin-up deviation). The heating effect appears only at the spin-down stage, when the temperature of the star is low enough. The same process happens in the pure QS. However, in the HS, the roto-chemical heating effects appear in the accretion phase because chemical deviation and heating effect are large enough compared to the temperature at that time. The surface temperature enhances quickly and lasts about $10^{8.68}$ s $\simeq$ 15.2 yr; then, the heating effect decreases, which causes a fast decline of the surface temperature. The heating platform in the accretion phase is related to the accretion disk mass (accretion rate), larger accretion rate leads to larger chemical deviation, higher and longer heating platform, and earlier appearance of heating effect. For the heating effect at the spin-down stage, the heating features of HS are the same as pure NS.

## 4. Discussion and Conclusions

In the present work, we study the effects of the roto-chemical heating with fall-back disk accretion in the QS and HS, and demonstrate the difference between them and the pure NS. The chemical and temperature evolutions in the stars containing quark matter (QS and HS) have some special features. Firstly, there are two jumps in the chemical evolution of the pure QS, but the chemical evolutions of the HS and NS change smoothly. The faster Durca reaction rate in the quark matter makes the chemical deviation recover to the balance much easier, which leads to an abrupt change in the chemical evolution. Secondly, the roto-chemical heating effect appears both in the accretion phase and spin-down phase of the HS. The heating platform in the accretion phase relies on the accretion rate, larger accretion rate results in larger chemical deviation, higher and longer heating platform, and earlier appearance of heating effect.

Choosing reasonable parameters, the NS's theoretical results in spin period, magnetic field, and surface temperature accord well with the classical pulsar B0950+08 [15]. Although the temperature of the HS is lower than the NS, the temperature still can reach $10^{5.5}$ K at a late stage, which also can explain the high temperature behavior of old classical pulsar as PSR B0950+08 with reasonable parameters. The temperature of the QS is too low for B0950+08 but is reasonable for the PSR J2144-3933 with the surface temperature of $T < 42,000$ K. Comparing the observations of PSR J2144-3933 with our models of accretion roto-chemical heating, the theory temperature of the NS and HS looks higher than the observation. However, the surface temperature of the QS is $10^{3.8}$–$10^{4.8}$ K around $3 \times 10^8$ yr. Choosing suitable accretion parameters, the quark star model probably explains the observation of J2144-3933. This means that the QS with accretion roto-chemical heating is a latent candidate for the explanation of old pulsars with lower surface temperature.The accretion roto-chemical heating model isn't excluded by the thermal emission observations of old classical pulsars.

The NS in Cassiopeia A (Cas A) is one of the most important isolated NSs, whose age and temperature are well estimated: $t \approx 330 \pm 20$ yr [25] and $T_s \sim 2 \times 10^6$ K [26], which can test the thermal evolution theory of the NS. Furthermore, a steady decline of $T_s$ by about 4% was found [27] and confirmed by Shternin et al. [28]. According to the latest Chandra observations of Cas A, the resulting 13 temperature measurements over more than 18 years yield a ten-year cooling rate of $\approx$ 2% [29]. Page et al. [30] and Sheternin et al. [28] suggested that this rapid cooling is triggered by breaking and formation of Cooper pairs (PBF) process. Other possibilities include magnetic field decay [31], neutrino cooling induced fast rotation [32,33], slow thermal relaxation [34,35], and stellar fluid oscillations [36]. Here, we provide an another possibility that the rapid cooling of the NS is the disappearance of roto-chemical heating effect in the accretion phase of the HS. The heating effect in the accretion phase of the HS enhances the surface temperature to about $10^{6.5}$ K. With the disappearance of the heating effect, the surface temperature can drop from $10^6$ K to $10^{5.95}$ K in $10^{8.5}$ s $\simeq$ 10 yrs (as model 1), which is close to the cooling rate of the NS in the Cas A. This short time-scale cooling process sensitively depends on the accretion rate. Choosing reasonable accretion parameters and the HS model, the fast cooling caused by the disappearance of roto-chemical heating in the accretion phase may explain the short time-scale cooling trend of the star in Cas A too.

The quark matter in the neutron stars possibly forms a color superconductor such as two-color-superconducting (2SC) or a color-flavor-locked (CFL) phase. The neutrino reactions involving superconductivity particles are slowed, which makes deviation of chemical potential rise faster, and allows a larger fraction of the rotational energy to be stored as chemical energy. Meanwhile, superconductivity decreases the specific heat of the star too. The effects of roto-chemical heating are more dramatic in the star containing quark matter in the superconductivity phase. The surface temperature may be enhanced about $10^{0.3}$ K compared to the normal matter case, which is favorable for the hybrid star's explanation of classical PSR B0950+08 with high temperature. Furthermore, the fast cooling process due to the disappearance of heating effect in the hybrid star still exists, and the cooling rate isn't affected obviously by the superconductivity of quark matter. With suitable accretion parameters, the hybrid star with superconductivity matter is still a possible candidate for the Cas A. However, the surface temperature of the quark star with superconductivity core is still low for the PSR B0950+08. The detailed evolution of the star with superconductivity matter is worth pursuing in the future.

In our model, the accretion driven crustal heating by pycno-nuclear processes is not taken into account, which provides efficient heating process ($L_q \approx 10^{31}$–$10^{33} ergs^{-1}$) in the neutron star's crust. It is a quite efficient internal heating process. In our accretion roto-chemical heating model, the accretion process ceases in $10^5$–$10^6$ s after birth. Thus, the deep-crustal heating doesn't have an obvious heating effect at the time the roto-chemical heating activates, which is around 0.1–1 Myr. For the explanations of the old pulsars ($t > 10^7$ yr) such as PSR B0950+05 or PSR J2144-3933, even the younger neutron star in Cas A with the estimated age of $330 \pm 20$ yr, the deep-crustal heating isn't dominant and could be ignored in the accretion roto-chemical heating model.

Our study is based on a uniform star model, which provides a rough understanding of the roto-chemical heating effect with an accretion process in the stars containing quark matter. The results are very close to those for hybrid stars containing large mixed-phase cores if their mean density equals the assumed uniform density. Considering the structure of the star with realistic dense matter equation of states in the frame of general relativity, a more realistic cooling evolution model can give more chances to compare with the observations, such as fast cooling of the NS in Cas A and old classical pulsars such as PSR B0950+08 and PSR J2144-3933. In addition, the description of magnetic field evolution in this work is the fit results with a phenomenological law as Wei (2018) used, which is based on the calculations of magnetic field evolution. 2D or 3D magneto-thermal modeling will provide more details for our accretion roto-chemical heating. All of these works are valuable for the future.

**Author Contributions:** Methodology, X.-Y.X.; software, K.-T.W.; data curation, X.-H.M.; writing, W.W.; supervision, W.W. All authors have read and agreed to the published version of the manuscript.

**Funding:** This research was funded by National Natural Science Foundation of China with Grant No. 11903013.

**Acknowledgments:** We would like to thank Xiao-Ping Zheng, Xia Zhou, and Quan Chen for valuable discussions.

**Conflicts of Interest:** The authors declare no conflict of interest.

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
