# Peer review of "Roto-Chemical Heating with Fall-Back Disk Accretion in the Neutron Stars Containing Quark Matter"

_universe, doi:10.3390/universe6050062_

Round 1

Reviewer 1 Report

The authors study changes in the thermal evolution of neutron stars, hybrid stars, and quark stars caused by the fall-back of matter onto such objects. This fall-back triggers changes in the chemical equilibrium compositions of these objects, which, in turn, modify their energy balances and hence their temperature evolutions. The study is interesting and the results should be of interest to the broader nuclear/astrophysics community. Before I feel confident recommending this paper for publication, however, several improvements and
clarifications are in order.

1. A brief discussion of the crust should be added to the paper. In particular, which model has been used, which nucleonic compositions are assumed, is superfluidity of neutrons taken into account or not?

2. Accreting quark stars, even if bare initially, may build up a nuclear crusts and thus be envelope by ordinary nuclear crust matter. This needs to be discussed. More than that, however, how would this feature change the results shown in this paper?

3. What if quark matter forms a color superconductor? How would this phase and the most prominent condensation patterns (2SC, CFL) associated with such matter change the findings presented in this paper?

4. There are several excellent topical reviews (all accessible through the preprint archive) on neutron stars, quark matter, and quark stars. None of them is listed!

5. Add references at the end of lines 55, 61, 62, 72.

6. Several references (like Reisenegger1995, Wei2018, etc) need to be replaced with their corresponding numerical labels (i.e. [2] for Reisenegger1995).

7. What are the units if R_{d,10} in Table 1?. Same for P_0 in Table 1.

8. Is the angular frequency is rad/sec or 1/sec = Hz?

9. Fig. 1-4: "omig" seem to be mislabeled?

10. Refs. [7] and [23] are incomplete.

11. Ref. [26] has been published by now. The reference is ApJ 874:175, 2019.

Author Response

We thank the referee for the thorough review of our manuscript, we will address the concerns of referee and changes we applied as follows.

1.A brief discussion of the crust should be added to the paper. In particular, which model has been used, which nucleonic compositions are assumed, is superfluidity of neutrons taken into account or not?

The thermal structures of the crust and core evolve essentially independently during the epoch of neutrino cooling. In this approximation the core is assumed to be isothermal and to contain all of the star’s mass and thermal energy. The temperature of the core is regulated by the loss of neutrinos which do not interact with the crust. The surrounding crust acts as a thin insulating envelope with no sources or sinks of energy and contains all of the temperature gradient. Thus, for a given core temperature, the heat flux through the crust is constant, and the effective surface temperature is determined entirely by processes within the crust. As a good approximation one can treat matter in the crust as a two-component plasma with one species, the ions (consisting of nuclei and some bound electrons), immersed in a uniform neutralizing background of free electrons. No other particles (except photons) are present. We have generally assumed that for densities less than 6.6 x 106 g cm-3 the nucleus present is 56Fe. Superfluidity of neutrons isn’t taken into account.

We have added some discussions on the crust in the section 3 (line83-94) of revised version.

2.Accreting quark stars, even if bare initially, may build up a nuclear crust and thus be envelope by ordinary nuclear crust matter. This needs to be discussed. More than that, however, how would this feature change the results shown in this paper?

As the referee pointed out, accreting quark stars may be bare initially and build up a nuclear crust after accretion. In our calculations, we didn’t consider the formation process of crust, because the formation of crust is of short time scale, comparing to long time scale of the thermal evolution process. So we assumed that the quark stars, hybrid stars and neutron stars already have the same crust configuration and used the same equation to describe the relation between the interior temperature and surface temperature. It is reasonable for us to compare heating effects in different star models.

We have added some discussions on this issue in the section 3 (line99-104) of revised version.

3.What if quark matter forms a color superconductor? How would this phase and the most prominent condensation patterns (2SC, CFL) associated with such matter change the findings presented in this paper?

As the referee pointed out, if quark matter forms a color superconductor as 2SC or CFL phase, the neutrino reactions involving superconductivity particles are slowed, which makes deviation of chemical potential rise faster, and allows a larger fraction of the rotational energy to be stored as chemical energy. Meanwhile superconductivity decreases the specific heat of the star too. The effects of rotochemical heating are more dramatic in the star containing quark matter in superconductivity phase. The surface temperature of the star may be enhanced about 100.3K comparing to the normal matter case, which is favorable for the hybrid star’s explanation of classical pulsar B0950+08 with high temperature. Furthermore, the fast cooling process due to the disappearance of heating effect in the hybrid star still exists, and the cooling rate isn’t affected obviously by the superconductivity of quark matter. With suitable accretion parameters, the hybrid star with superconductivity matter is still possible candidate for the Cas A. But the surface temperature of the quark star with superconductivity core is still low for the pulsar B0950+08. The detailed evolution of the star with superconductivity matter is worth pursuing in the future.

We have added some discussions in the section 4 (line191-204) of revised version.

4.There are several excellent topical reviews (all accessible through the preprint archive) on neutron stars, quark matter, and quark stars. None of them is listed!

Thank referee’s suggestion, we have added the following references in revised version.

Tsuruta, S., Phys.Rep., 292(1), 1998.

Page, D., Lattimer,J.M., Prakash,M. & Steiner,A.W., ApJS, 155(623), 2004.

Madsen, J., Physics and astrophysics of strange quark matter, Hadrons in Dense Matter and Hadrosynthesis, 516, 1999

J.M. Lattimer & M.Prakash, Science, 304(536), 2004

Yakovlev, Dmitrii G., Levenfish, Kseniya P., Shibanov, Yurii A., Physics Uspekhi, 42(737) ,1999

Weber, F. Progress in Particle and Nuclear Physics, 54(193), 2005

5.Add references at the end of lines 55, 61, 62, 72.

We have added the references in revised version.

6.Several references (like Reisenegger1995, Wei2018, etc) need to be replaced with their corresponding numerical labels (i.e. [2] for Reisenegger1995).

Sorry for our carelessness, we have revised the reference.

7.What are the units if R_{d,10} in Table 1?. Same for P_0 in Table 1.

The unit of Rd,10 is 1010 cm and the unit of P0 is ms, we have added the units in Table 1.

8.Is the angular frequency is rad/sec or 1/sec = Hz?

The unit of angular frequency is rad/sec and we have revised the unit in Fig1-4.

9.Fig. 1-4: "omig" seem to be mislabeled?

We have changed “omig” to “Ω”in Fig.1-4.

10.Refs. [7] and [23] are incomplete.

We have completed the references.

11.Ref. [26] has been published by now. The reference is ApJ 874:175, 2019.

We have revised the reference.

Thank referee for so careful review of our manuscript, all changes are marked in bold in the revised version.

Reviewer 2 Report

Referee review for the article "Roto-chemical heating with fall-back disk accretion in the neutron stars containing quark matter"
by Wei Wei, Xin-Yu Xu, Kai-Tuo Wang and Xiao-Hang Ma.

In this work the authors present results for processes of heating of rotating compact stars with the inclusion of accretion from a fall-back disk.
In their study three classes of compact stars are included: pure hadronic stars, pure quark stars and hybrid stars composed of a quark matter core covered by a mantle of hadronic matter. In the calculations the effects of magnetic fields are considered. The critical parameters of the model are the  accretion disk mass M_d, initial magnetic strength B_0 and period P_0, magnetic decay parameter M_0 and initial disc radius R_d. The calculations are performed
for fourth set of parameters defining four models. By choosing reasonable parameter values, the authors are able to explain observations, like the abrupt change in the Temperature of CAS A.

In my opinion the article is well presented and well explained and has results worth of publication, after some minor corrections. Therefore, I would like to ask the authors to clarify a couple of details and moreover I would like to point out some minimal typo corrections.

What is the mass of the uniform density compact star considered in your calculations? Does it lead to some reasonable value for the radius of the star? what is it?

Why do the upper panels of the figures corresponding to surface magnetic field and spin frequency only show a black line instead of three (red, blue, back) ?

I found some typos:

Line 43:

Heating maybe provide -> may provide...

Line 65 and 66:

Other symbols refer to the reference ->

For the other quantities we refer to the reference...

Similarly,

Line 76:

The spin and magnetic evolution equations refer to ->

For the spin and magnetic evolution equations we refer to...

In all the figures caption: the correct order of units is K, K, G and s^-1, respectively.

Line 135: more easier -> much easier

Author Response

We thank the referee for the thorough review of our manuscript, we will address the concern of referee and changes we applied as follows.

1.What is the mass of the uniform density compact star considered in your calculations? Does it lead to some reasonable value for the radius of the star? what is it?

For the uniform density star model used in our work and previous studies [Reisenegger1995, Cheng1996], the chemical and thermal evolution results don’t depend on the mass of star, which means our calculation results apply to all stars with reasonable mass and same average density. The baryon number density we used in our calculation is n = 0.4fm-3, which is a reasonable average density for neutron stars and ensures reasonable star mass and radius.

We have added some explanations in the section 3 (line81-82) of revised version.

2.Why do the upper panels of the figures corresponding to surface magnetic field and spin frequency only show a black line instead of three (red, blue, back) ?

Figure 1-4 are evolution curves of three stellar models corresponding to 4 sets of parameters showed in Table 1. Each figure corresponds to one set of parameters, which means accretion processes for three stars are same in each figure. The evolutions of spin frequency and surface magnetic field depend on the accretion parameters and process, which aren’t related to the inner matter of star in our calculations. But the surface temperature and chemical deviation are related to the star model. So spin angular frequency and surface magnetic field curves of three stars in each figure overlap to each other.

We have added some explanations in the section 3 (line110-115) of revised version.

3.I found some typos:

Line 43:Heating maybe provide -> may provide...

Line 65 and 66:Other symbols refer to the reference ->For the other quantities we refer to the reference...Similarly,

Line 76:The spin and magnetic evolution equations refer to ->For the spin and magnetic evolution equations we refer to...

In all the figures caption: the correct order of units is K, K, G and s^-1, respectively.

Line 135: more easier -> much easier

Thank referee’s careful review, we have revised the typos. All changes are marked in bold in the revised version.

Round 2

Reviewer 1 Report

The authors have modified the manuscript according to the suggestions
of my previous report. The article is now suitable for publication.